# RELITTRELLIS: LIGHTING-HOMOGENIZED STRUCTURED 3D LATENTS FOR RELIGHTABLE 3D GENERATION

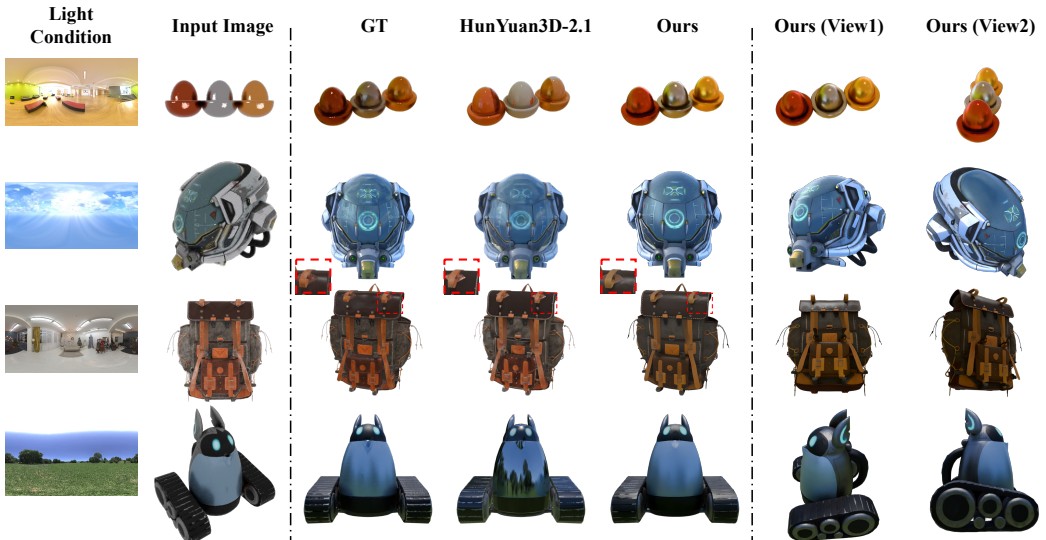

Figure 1: Relightable 3D generation results. From left to right: target lighting conditions, input images, ground truth, HuanYuan3D-2.1 results, and our results (including two novel views). Our method achieves more accurate relighting compared to existing approaches, particularly in preserving material properties and lighting consistency across views.

## ABSTRACT

Generating relightable 3D assets from a single image is fundamentally ill-posed: geometry, material, and lighting are deeply entangled, making both principle-driven decomposition and end-to-end neural generation brittle or inconsistent. We propose **RelitTrellis**, a homogenize-then-synthesize framework built on a **Lighting-Homogenized Structured 3D Latent (LH-SLAT)**. LH-SLAT attenuates shadows and unstable highlights while preserving geometry-consistent diffuse cues, providing a well-conditioned substrate for relighting. From a casually lit input, RelitTrellis first derives LH-SLAT and then synthesizes 3D Gaussian parameters conditioned on target illumination, efficiently capturing higher-order light–material interactions such as soft shadows and indirect reflections. Experiments on Digital Twin Category, Aria Digital Twin, and Objaverse benchmarks show that RelitTrellis achieves state-of-the-art quality, strong cross-object and cross-illumination generalization, consistent multi-view rendering, and real-time feed-forward inference without per-object optimization.

## 1 INTRODUCTION

Generating relightable 3D assets from a single image is a challenging problem in vision and graphics, with applications in virtual commerce, XR, and digital twins. The difficulty stems from en-

tangled geometry, material, and lighting in a single RGB observation, which makes standard PBR decomposition brittle and end-to-end neural relighting inconsistent.

The first is principle-driven rendering based on physically-based rendering (PBR) decomposition Zhao et al. (2025); Liang et al. (2025). These methods follow a decomposition pipeline: regress albedo, roughness, and metallic maps, then re-render under novel illumination. While interpretable and editable, the inverse problem is highly ill-posed: real images embed diffuse shading, cast shadows, inter-reflections, and view-dependent highlights that do not map cleanly into isotropic BRDFs. Small estimation errors—such as mistaking shadows for low albedo—are amplified nonlinearly during re-rendering, leading to baked-in shadows or distorted reflections.

The second is data-driven rendering, which leverages large generative models to directly map images to relighted results Jin et al. (2024); Zhang et al. (2025). Such models capture complex global light transport and can synthesize visually compelling outputs. However, they face critical limitations: representations are black-box with little controllability, training requires massive paired relighting data that are scarce in practice, diffusion-based inference is slow and stochastic, and multi-view consistency remains unresolved.

Both families struggle with a common barrier: the input image itself is imprinted with arbitrary, scene-specific illumination that entangles material and lighting, complicating consistent relighting. To address this, we argue that before decomposition or generation, a crucial step is to homogenize illumination. The goal is to map a casually lit observation into a canonical representation in which shadows and unstable specularities are attenuated while geometry-aligned diffuse cues remain. From the perspective of problem formulation, this step fundamentally differs from the ill-posed inverse estimation in PBR decomposition: instead of disentangling multiple entangled factors at once, light homogenization provides a more stable and well-conditioned transformation. At the same time, compared to purely generative mappings, introducing an intermediate structured latent enhances controllability and interpretability, offering a principled substrate for subsequent synthesis. Inspired by the principle of Vasluianu et al. (2024), we introduce a Lighting-Homogenized Structured 3D Latent (LH-SLAT), which suppresses shadows and unstable highlights while preserving geometry-consistent diffuse cues essential for faithful relighting as shown in Figure 2.

Building on LH-SLAT, we propose RelitTrellis, a homogenize-then-synthesize framework for single-image relightable 3D asset generation. From a casually lit input, RelitTrellis first extracts a canonical structured latent under homogenized illumination. This removes scene-specific illumination patterns that traditionally hinder material–lighting separation. Then, a lightweight decoder conditions on target environment lighting and a dedicated light-ray encoding to synthesize 3D Gaussian parameters and appearance, efficiently capturing higher-order light–material interactions such as soft shadows and indirect reflections.

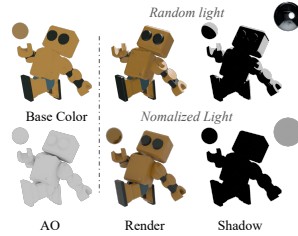

Figure 2: Visualization of renders under random lighting and homogenized illumination.

This design enables a maximized setup: unlike prior neural-based rendering approaches that suffer from slow and stochastic diffusion sampling and multi-view inconsistency, our homogenize-then-synthesize pipeline scales to diverse inputs, generalizes across objects and lighting, and achieves real-time inference without per-object optimization.

Experiments on various datasets confirm that RelitTrellis achieves state-of-the-art quality with strong generalization across objects and lighting, faithful reproduction of specular and shadow effects, and real-time inference without per-object optimization.

Our contributions are:

1. We introduce a lighting-homogenized structured latent (LH-SLAT) that suppresses shadows and unstable highlights while retaining geometry-consistent diffuse cues.

2. We design a homogenize-then-synthesize pipeline that couples LH-SLAT extraction with light-conditioned synthesis of 3D Gaussian parameters and appearance.

3. We conduct extensive experiments and ablations on various datasets, demonstrating state-of-the-art or improved quality, cross-object and cross-illumination generalization, multi-

view consistency, and real-time feed-forward relighting (up to 48 FPS) without per-object tuning.

## 2 RELATED WORKS

### 2.1 IMAGE RELIGHTING AND INVERSE RENDERING

Image relighting and inverse rendering sit at the intersection of geometry, material estimation, and light transport, and have been studied from both physics-driven and data-driven perspectives Jin et al. (2024). Classical inverse rendering methods (e.g., SIRFS) recover interpretable PBR maps (albedo, roughness, normals) via optimization and hand-crafted priors Barron & Malik (2013). These pipelines are interpretable and editable, but the inverse problem is highly ill-posed in real scenes: shadows, inter-reflections and view-dependent highlights easily bias the recovered materials and produce baked-in artifacts under re-rendering (see Sec. 1).

Recent learning-based relighting approaches fall into two broad families. The first family focuses on physically-structured decomposition into PBR components and subsequent re-rendering Zhao et al. (2025); Liang et al. (2025). Decomposition-based methods are well suited for relightable asset creation because they produce interpretable, editable material maps; however, the inverse problem is ill-posed from casual single-view inputs and robustness often requires multi-view data or per-object optimization.

The second family targets direct, often diffusion-based, image relighting and editing: methods such as SPOTLIGHT, DiLightNet, IC-Light and LightLab exploit the generative power of diffusion priors to produce high-fidelity relit images and offer fine-grained light control Fortier-Chouinard et al. (2024); Zeng et al. (2024a); Zhang et al. (2025); Magar et al. (2025). While visually compelling, these approaches are typically computationally expensive, stochastic at inference, require large paired data, and do not naturally provide multi-view consistent 3D assets.

In this work we take a middle path: instead of directly solving a brittle PBR inversion or relying on black-box diffusion sampling, we first homogenize the input illumination into a canonical representation (LH-SLAT) and then synthesize a relightable 3D field in a feed-forward manner (Sec. 2). This homogenize-then-synthesize strategy stabilizes downstream decoding and improves controllability.

### 2.2 DIFFUSION PRIORS AND 3D CONTENT GENERATION

Diffusion priors and score-distillation techniques have catalyzed rapid progress in 3D synthesis from 2D models Poole et al. (2022); Tang et al. (2023); Shi et al. (2023). DreamFusion and follow-up works transfer 2D generative knowledge to 3D via SDS, improving fidelity at the cost of expensive iterative optimization. More recent efforts push for faster or feed-forward 3D reconstruction and native 3D generation by training on 3D datasets, or by designing architectures that decouple geometry and appearance Hong et al. (2023); Xiang et al. (2024); Zhang et al. (2024a). These native or decoder-first approaches tend to provide better cross-view consistency and faster inference than optimization-based SDS pipelines, but aligning geometry and high-fidelity appearance remains challenging. Our method builds on this line: we adopt a structured 3D latent representation and a decoder that directly predicts a relightable 3D Gaussian Splatting (3DGS) field, trading expensive optimization for a real-time, multi-view-consistent synthesis.

### 2.3 RELIGHTABLE 3D ASSET SYNTHESIS

Producing relightable 3D assets requires models to represent both intrinsic surface properties and lighting-dependent transport (shadows, speculars, interreflections). Prior works condition NeRFs, Gaussian splats or meshes on lighting inputs to enable relighting-aware outputs Zeng et al. (2023); Jin et al. (2024); Li et al. (2023); Gao et al. (2024); Bi et al. (2024). Many approaches either use volumetric neural renderers that are costly at inference, or attempt to estimate PBR maps without lighting supervision, which leads to poor disentanglement Qiu et al. (2024); Liu et al. (2024); Shim et al. (2024). Recent models explore large inverse-rendering architectures to predict PBR properties from sparse views, but computational cost and per-object optimization remain bottlenecks Li et al. (2025b); Zhang et al. (2024b).

Figure 3: Pipeline of **RelitTrellis**. Stage 1: *Light Homogenization* extracts a Lighting-Homogenized Structured 3D Latent (**LH-SLAT**) from a casually lit input image. Stage 2: *Relightable 3DGS Syntehsis* generates a relightable 3D Gaussian Splatting (**3DGS**) field conditioned on the LH-SLAT, target illumination, and target viewpoint. The decoded 3DGS encodes geometry, appearance, and light–material interactions, and is rendered into the final relit image.

In contrast, our homogenize-then-synthesize pipeline explicitly removes unstable, scene-specific illumination before decoding. This reduces the ill-posedness of PBR inversion and enables a single feed-forward decoder to produce relightable 3DGS with real-time rendering and improved multi-view consistency (see Sec. 1 and Sec. 3). The design occupies an intermediate point between interpretable PBR pipelines and powerful but costly diffusion-based renderers, combining stability, controllability, and practical speed for relightable asset creation.

## 3 PRELIMINARY

**3D Gaussian Splatting (3DGS).** 3DGS Kerbl et al. (2023) represents a breakthrough in neural rendering by employing anisotropic 3D Gaussians as explicit scene representations. Each Gaussian is parameterized by its center $x \in \mathbb{R}^3$, opacity $\sigma \in [0, 1]$, and covariance $\Sigma \in \mathbb{R}^{3\times3}$, which is decomposed into a rotation quaternion $r$ and scaling vector $s$:

$$\Sigma = RSS^T R^T. \tag{1}$$

For rendering, Gaussians are projected to 2D via the covariance transformation:

$$\Sigma' = JV\Sigma V^T J^T, \tag{2}$$

where $J$ is the Jacobian of projection and $V$ is the view matrix. Pixel color is computed via alpha blending:

$$C(\mathbf{p}) = \sum_{i \in N} T_i \alpha_i c_i, \quad \alpha_i = \sigma_i e^{-\frac{1}{2}(\mathbf{p}-\mu_i)^T \Sigma'^{-1}(\mathbf{p}-\mu_i)}, \tag{3}$$

where $\mathbf{p}$ is the pixel coordinate, $\mu_i$ is the projected Gaussian center, $N$ is the ordered list of Gaussians intersecting the ray, and $T_i = \prod_{j=1}^{i-1}(1 - \alpha_j)$ is the transmittance. This formulation enables differentiable, real-time rendering.

**Structured 3D Latents (SLAT).** SLAT Xiang et al. (2024) provides a compact yet expressive representation for 3D content generation. Unlike dense voxels, SLAT encodes only surface-adjacent regions. We denote the SLAT collection by $Z = \{(z_i, p_i)\}_{i=1}^L$, where each token feature $z_i \in \mathbb{R}^C$ is associated with a position $p_i \in \{0, 1, \ldots, N-1\}^3$, and $L \ll N^3$. A decoder $\mathcal{D}$ can generate 3DGS, radiance fields, or meshes from the same SLAT, making it well-suited for relighting tasks.

## 4 METHOD

The challenge in single-image 3D relighting lies in disentangling lighting from intrinsic object properties, since shadows, highlights, and interreflections entangle with geometry. To avoid unstable PBR inversion and black-box neural generation, we propose a **homogenize-then-synthesize** framework: first extract a Lighting-Homogenized SLAT (LH-SLAT), then decode a relightable 3DGS.

Our framework consists of two stages:

**Stage 1: Light Homogenization — LH-SLAT Extraction.** A rectified flow model $f_\theta$ maps the casually lit input image $I_{\text{in}}$ into a lighting-homogenized latent $Z_{\text{lh}}$:

$$Z_{\text{lh}} = f_\theta(I_{\text{in}}). \tag{4}$$

$Z_{\text{lh}}$ suppresses unstable shadows and highlights while preserving geometry-consistent cues.

**Stage 2: Relightable 3DGS Synthesis.** A feed-forward decoder $\mathcal{D}$ generates a relightable Gaussian field $\mathcal{G}$ conditioned on $Z_{\text{lh}}$, the target view $\mathbf{v}_{\text{target}}$, and target illumination $L_{\text{target}}$ encoded by $\mathcal{E}_l$:

$$\mathcal{G} = \mathcal{D}\big(Z_{\text{lh}}, \mathbf{v}_{\text{target}}, \mathcal{E}_l(L_{\text{target}})\big). \tag{5}$$

The final relighted image is rendered by a differentiable rasterizer $\mathcal{M}$:

$$I_{\text{target}} = \mathcal{M}(\mathcal{G}, \mathbf{v}_{\text{target}}). \tag{6}$$

In the following, we describe Stage 1 (Sec. 4.1) and Stage 2 (Sec. 4.2) in detail.

### 4.1 LH-SLAT Extraction & Generation

The first stage generates a Lighting-Homogenized Structured 3D Latent (LH-SLAT) $Z_{\text{lh}}$ from a single input image $I_{\text{in}}$ captured under unknown illumination. LH-SLAT serves as a stable substrate for downstream synthesis.

**Lighting Homogenization.** We define the homogenized light as a uniform, white ambient environment illumination. This eliminates hard shadows and promotes a more uniform distribution of diffuse and specular reflection energy (Figure 2). We extract SLAT features under this lighting to create an intermediate representation suitable for relighting.

**LH-SLAT Extraction.** To train $f_\theta$, we prepare paired data $(I_{\text{in}}, Z_{\text{lh}})$ via multi-step rendering of 3D assets under homogenized lighting. As shown in 4, we first generate the ground-truth homogenized latents $Z_{\text{lh}}$: (1) for each 3D asset, we render $N$ views under our defined homogenized illumination; (2) we extract dense 2D visual features using a pre-trained DINOv2 model; (3) these features are back-projected into a sparse 3D voxel grid; (4) finally, this sparse grid is compressed by a pre-trained SLAT VAE encoder to obtain $Z_{\text{lh}}$. Second, to create the corresponding input $I_{\text{in}}$, we render $M$ additional images of the same asset under diverse, random lighting conditions and camera poses.

Optionally, for highly reflective materials, we extract Basecolor SLAT $Z_{\text{bc}}$ from multi-view basecolor renderings, concatenating with $Z_{\text{lh}}$ to retain base color information.

**LH-SLAT Generation.** As shown in 4, we use a rectified flow model $f_\theta$ to generate the lighting-homogenized SLAT $Z_{\text{lh}}$ from the input image $I_{\text{in}}$. The rectified flow model is trained to learn the mapping between the arbitrarily lit image and the corresponding latent representation under our homogenized lighting conditions. Specifically, we fine-tune a pre-trained SLAT rectified flow model Xiang et al. (2024) using LoRA Hu et al. (2022). The loss function for training is the conditional flow matching loss $\mathcal{L}_{stage1}$:

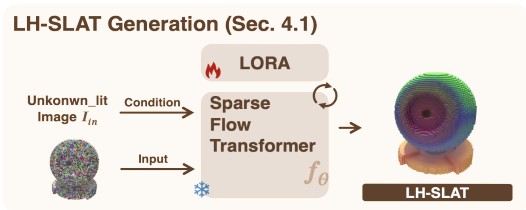

**LH-SLAT Generation (Sec. 4.1)**

Figure 4: The pipeline for LH-SLAT generation.

$$\mathcal{L}_{stage1} = \mathbb{E}_{t, \mathbf{z}_0, \boldsymbol{\epsilon}} \|\mathbf{v}_\theta(\mathbf{z}, t) - (\boldsymbol{\epsilon} - \mathbf{z}_0)\|_2^2, \tag{7}$$

where $\mathbf{z}(t) = (1-t)\mathbf{z}_0 + t\boldsymbol{\epsilon}$ is the linear interpolation between the data sample $\mathbf{z}_0$ and noise $\boldsymbol{\epsilon}$, and $\mathbf{v}_\theta$ approximates the time-dependent vector field. If the optional basecolor SLAT $\mathbf{z}_{\text{bc}}$ is used, it is concatenated with $\mathbf{z}_{\text{lh}}$ to provide additional color information to the subsequent stage.

### 4.2 Relightable 3DGS Synthesis

The second stage synthesizes a relightable 3D Gaussian Splatting (3DGS) field $\mathcal{G}$ from LH-SLAT, conditioned on target illumination and viewpoint. Unlike iterative optimization approaches Gao et al. (2024); Bi et al. (2024), we employ an efficient feed-forward decoder with two key modules: the *Intrinsic Aware Decoder (IAD)* and the *Environment Aware Renderer (EAR)*.

### 4.2.1 INTRINSIC AWARE DECODER (IAD)

The goal of IAD is to process latent representations $Z_{lh}$ and generate a view-independent and illumination-invariant intrinsic feature $\boldsymbol{h} = \{(\boldsymbol{h}_i, \boldsymbol{p}_i)\}_{i=1}^{L}$, where $\boldsymbol{h}_i \in \mathbb{R}^{768}$. This sparse feature field $\boldsymbol{h}$ effectively decodes the underlying geometric structure and material properties of the scene. To achieve this, IAD employs a Transformer architecture akin to TRELLIS Xiang et al. (2024), leveraging stacked self-shifted window attention blocks to exploit the inherent locality of structured 3D latent sequences. To further enhance the model's comprehension of global structural relationships and lighting context, a register cross-attention layer is incorporated into each block. Specifically, 16 learnable register tokens are appended to each object's corresponding SLAT token sequences. These tokens encode global scene information and potentially attenuate high-frequency noise within the embeddings Darcet et al. (2024); Li et al. (2025a). Finally, these register tokens are injected into the decoder via a global cross-attention mechanism, facilitating information exchange between the register tokens and all latent variable tokens, thereby enabling the generation of a coherent and globally consistent intrinsic representation.

### 4.2.2 ENVIRONMENT AWARE RENDER (EAR)

EAR receives the intrinsic feature $\boldsymbol{h}$ and synthesizes 3D Gaussian Splatting (3DGS) parameters $\mathcal{G}$ by incorporating view embeddings and light conditions, as illustrated in Figure 3.

**Observe view embedding**. Since specular highlights vary under different viewing angles, we abandon the commonly used spherical harmonics and instead inject the observed view information into the learning process of EAR from the outset to enhance the model's perception of specular highlights. Along the camera ray to each voxel $\boldsymbol{p}_i$ in the world coordinate system, we record the distance $x = \{(l_i, \boldsymbol{p}_i)\}_{i=1}^{L}$, where $l_i \in \mathbb{R}$, and the ray direction $\boldsymbol{d}^w = \{(\boldsymbol{d}^w_i, \boldsymbol{p}_i)\}_{i=1}^{L}$. We then transform $\boldsymbol{d}^w$ to the camera coordinate system using the extrinsic matrix, denoted as $\boldsymbol{d} = \{(\boldsymbol{d}_i, \boldsymbol{p}_i)\}_{i=1}^{L}$, where $\boldsymbol{d}_i \in \mathbb{R}^3$. We apply NeRF positional encoding and learnable positional encoding to $\boldsymbol{d}$ and $l$ voxel-wise, respectively, ultimately obtaining the view embedding

$$\boldsymbol{e}^v = \{\boldsymbol{e}^d, \boldsymbol{e}^l\} = \{(\boldsymbol{e}_i^d, \boldsymbol{p}_i), (\boldsymbol{e}_i^l, \boldsymbol{p}_i)\}_{i=1}^{L}, \quad \boldsymbol{e}_i \in \mathbb{R}^{768}.$$

Then, we add $\boldsymbol{e}^d$ and $\boldsymbol{e}^l$ voxel-wise to $\boldsymbol{h}$ to obtain $\boldsymbol{h}^v$, which serves as the input to EAR.

**HDR lighting condition**. We encode the environment map $\mathbf{E}$ as lighting conditions using an HDRI encoder $\mathcal{E}_l$. Similar to previous works Jin et al. (2024); Liang et al. (2025); He et al. (2024), we obtain the low dynamic range (LDR) image $\mathbf{E}_{ldr}$ through Reinhard tone mapping, compute the normalized log-intensity map $\mathbf{E}_{log} = \log(\mathbf{E} + 1)/\mathbf{E}_{max}$, and generate the direction encoding $\mathbf{E}_{dir} \in \mathbb{R}^{H \times W \times 3}$ in the camera coordinate system. Differently from using a VAE encoder to compress $\mathbf{E}$, we employ ConvNeXt to extract multi-scale visual features from the LDR image $\mathbf{E}_{ldr}$ and the normalized log-intensity map $\mathbf{E}_{log}$. A key innovation is that we avoid directly compressing the direction encoding $\mathbf{E}_{dir}$. Instead, we first encode it through (NeRF) position embedding and then fuse it with visual features at multiple scales using the **Spatial Cross Attention**. The spatial cross attention acts as a learnable positional encoding, modulating the visual features at different scales via $\mathbf{E}_{dir}$ and embedding directional information into the visual representation. These multi-scale features are then concatenated along the channel dimension, further processed with positional encoding, and passed through three self-attention blocks to form the corresponding lighting condition $C_L \in \mathbb{R}^{4096 \times 768}$. This design allows us to edit $\mathbf{E}_{dir}$ when switching views and lighting directions.

EAR primarily consists of stacked cross-attention blocks. The lighting condition $C_L$ is injected into the intrinsic feature $\boldsymbol{h}^v$ via cross-attention layers, enabling the network to be aware of the environment lighting conditions. Similar to IAD, to enhance the perception of global illumination, we incorporate a register cross-attention layer in each block. After EAR, we obtain the lighting-aware sparse feature $\boldsymbol{h}^e$.

**3D Gaussian Decoding**. We utilize 3D GS as the final relighting representation. Specifically, after view encoding and EAR, we obtain the lighting-independent feature $\boldsymbol{h}^v$ and the lighting-dependent feature $\boldsymbol{h}^e$. The 3D GS decoding process can be represented as:

$$\{(\boldsymbol{h}_i^v, \boldsymbol{p}_i)\}_{i=1}^{L} \rightarrow \{\{(\boldsymbol{o}_i^k, \boldsymbol{b}_i^k, \gamma_i^k, \boldsymbol{m}_i^k, \boldsymbol{s}_i^k, \alpha_i^k, \boldsymbol{r}_i^k)\}_{k=1}^{K}\}_{i=1}^{L}, \{(\boldsymbol{h}_i^e, \boldsymbol{p}_i)\}_{i=1}^{L} \rightarrow \{\{(\boldsymbol{f}_i^k, \hat{s}_i^k, \sigma_i^k)\}_{k=1}^{K}\}_{i=1}^{L} \quad (8)$$

At each voxel location $\boldsymbol{p}_i$, we decode the intrinsic feature $\boldsymbol{h}_i^v$ into the parameters of $K$ Gaussians, including position offset $\boldsymbol{o}$, base color $\boldsymbol{b}$, roughness $\gamma$, metallic $\boldsymbol{m}$, scale $\boldsymbol{s}$, opacity $\alpha$, and rotation $\boldsymbol{r}$.

Table 1: Quantitative comparison against state-of-the-art methods across four sub-tasks.

| | ADT | | | DTC | | | Objaverse data | | | Glossy Synthetic dataset | | |
|---|---|---|---|---|---|---|---|---|---|---|---|---|
| | LPIPS↓ | PSNR↑ | SSIM↑ | LPIPS↓ | PSNR↑ | SSIM↑ | LPIPS↓ | PSNR↑ | SSIM↑ | LPIPS↓ | PSNR↑ | SSIM↑ |
| **G-Buffers Forward Rendering** | | | | | | | | | | | | |
| DiffusionRenderer | 0.0802 | 24.41 | 0.9172 | 0.0560 | 27.16 | 0.9354 | 0.0616 | 27.09 | 0.9288 | 0.0707 | 25.46 | 0.9126 |
| Ours | **0.0488** | **29.15** | **0.9484** | **0.0458** | **31.59** | **0.9586** | **0.0490** | **32.23** | **0.9627** | **0.0475** | **30.47** | **0.9594** |
| **Random-lit Single-image Reconstruction** | | | | | | | | | | | | |
| RGB↔X | 0.1605 | 15.15 | 0.8445 | 0.1349 | 15.48 | 0.8624 | 0.1199 | 16.09 | 0.8801 | 0.1271 | 14.29 | 0.8612 |
| DiLightNet | 0.0949 | 21.11 | 0.8947 | 0.0650 | 23.53 | 0.9147 | 0.0507 | 25.65 | **0.9300** | 0.0523 | 24.09 | 0.9213 |
| DiffusionRenderer | **0.0767** | 22.50 | **0.9105** | 0.0579 | 23.70 | **0.9234** | 0.0516 | 24.81 | 0.9285 | 0.0547 | 23.40 | 0.9163 |
| Ours | 0.0819 | **22.85** | 0.9006 | **0.0551** | **24.35** | 0.9095 | **0.0407** | **26.24** | 0.9252 | **0.0371** | **25.02** | **0.9224** |
| **Unknown-lit Single-image Relighting** | | | | | | | | | | | | |
| DiLightNet | 0.1037 | 20.59 | 0.8813 | 0.0729 | 22.63 | 0.8913 | 0.0657 | 23.87 | 0.9011 | 0.0622 | **22.40** | 0.9059 |
| NeuralGrafferer | 0.2675 | 14.31 | 0.7839 | 0.2548 | 14.22 | 0.7943 | 0.2108 | 14.68 | 0.8238 | 0.1767 | 15.67 | 0.8200 |
| DiffusionRenderer | **0.0916** | **21.91** | **0.8960** | 0.0691 | 22.99 | 0.9078 | 0.0609 | 23.75 | 0.9169 | 0.0632 | 22.13 | 0.9062 |
| Ours | 0.1020 | 21.75 | 0.8857 | **0.0664** | **23.12** | **0.9123** | **0.0587** | **23.96** | **0.9234** | **0.0486** | 22.19 | **0.9216** |
| **Novel-view Relighting** | | | | | | | | | | | | |
| 3DTopia-XL | 0.1754 | 17.24 | 0.8013 | 0.1051 | 21.56 | 0.8674 | 0.0769 | 23.22 | 0.8989 | 0.0857 | 20.89 | 0.8807 |
| Stable-Fast-3D | 0.1028 | 19.43 | 0.8881 | 0.0616 | 22.07 | 0.9154 | 0.0666 | 22.26 | 0.9112 | 0.0747 | 20.17 | 0.8943 |
| MeshGen | 0.0939 | 20.15 | 0.8879 | 0.0661 | 22.87 | 0.9101 | 0.0509 | 24.15 | 0.9306 | 0.0637 | 21.43 | 0.9071 |
| Hunyuan3D-2.1 | 0.0727 | 22.30 | **0.9017** | **0.0481** | 24.89 | 0.9255 | 0.0479 | 25.47 | 0.9328 | 0.0533 | 22.26 | **0.9119** |
| Ours | **0.0702** | **22.67** | 0.8983 | 0.0503 | **25.32** | **0.9278** | **0.0462** | **25.78** | **0.9379** | **0.0514** | **22.79** | 0.9088 |

For the opacity $\alpha$, we use a $\tanh$ activation function to support negative density and enhance expressive power Zhu et al. (2025). We use the lighting-dependent feature $\boldsymbol{h}_i^e$ to decode each Gaussian's 48-dimensional color feature $\boldsymbol{f}$, lighting-related scale $\hat{s}$ and shadow $\sigma$. The final center position of each Gaussian is $\boldsymbol{x}_i^k = \boldsymbol{p}_i + \tanh(\boldsymbol{o}_i^k)$. We calculate the shortest axis based on the scale $\hat{s}$, and use it as the normal vector for each Gaussian primitive. Finally, we employ a simple shallow MLP network that combines the positional encoding of the normal vector and the color feature $\boldsymbol{f}$. This network uses ReLU activation functions in its intermediate layers and an ELU activation function in its final layer to predict the radiance values for each Gaussian. Through the rasterization operation $\mathcal{M}$, we obtain the 2D HDR prediction $I_{target}^{hdr}$. We also render 2D base color, roughness, metallic, shadow images $I^b, I^r, I^m, I^s$.

**Loss Function**. We supervise the training by calculating the reconstruction loss $\mathcal{L}_{hdr}$ between the rendered reference HDR image and the predicted HDR result consists of $L_1$, LPIPS Zhang et al. (2018), and D-SSIM. Following Zeng et al. (2025), to prevent minor errors in the high-light areas from dominating the $L_1$ loss, we apply a logarithmic transformation to the images. Then, we compute the LPIPS and DSSIM losses on the tonemapped versions of the two images (clamp(log I / log 2, 0, 1)). Furthermore, we calculate $L_1$ losses between the rendered reference and predicted results for material properties (including base color, roughness, and metalness) and shadows to aid in training. The total loss is a weighted sum of each individual loss, represented as: $\mathcal{L}_{stage2} = \mathcal{L}_{hdr} + \lambda_{pbr}\mathcal{L}_{pbr} + \lambda_{shadow}\mathcal{L}_{shadow}$. For more details refer to the appendix.

## 5 EXPERIMENTS

### 5.1 IMPLEMENTATION DETAILS

More training details refer to the Appendix. A.1.

**Training data**. Our training dataset comprises 87K 3D assets with physically-based rendering (PBR) textures, curated from the Objaverse-XL dataset. These assets are illuminated using 2,000 High Dynamic Range Images (HDRIs), each at 4K resolution, used as environment maps. We normalized the assets to fit within a bounding box of $[-0.5, 0.5]$. The first training stage involves rendering 150 viewpoints under normalized lighting to extract illumination-invariant structural latent representations. For input images under unknown illumination, camera poses are sampled with yaw within ±45 degrees and pitch from -10 to 45 degrees, oriented towards the object's center, and with field of view (FOV) and radius following Xiang et al. (2024). Unknown illumination is modeled with (1) six area lights uniformly distributed on a sphere, (2) 1-3 area lights randomly sampled within the camera's hemisphere, or (3) a random, Z-axis-rotated environment map. Area light intensities are sampled uniformly between 300 and 700 (units), distances between 5 and 8 units. In the second stage, we re-light objects using randomly rotated environment maps, with a fixed FOV of 40

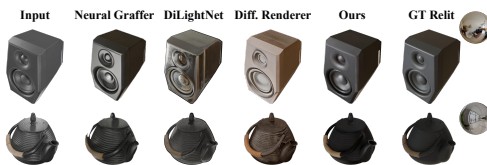

Figure 5: Visual comparison of Diffusion Renderer with Gbuffer/LH-SLAT for image relighting.

Figure 6: Visual comparison of image reconstruction.

Figure 7: Visual comparison of relighting.

degrees and camera positions uniformly sampled on a sphere of radius 2. Across both stages, each object is rendered from 12 viewpoints, each under 16 illuminations, using Blender EEVEE Next Team (2025).

**Task definitions and baselines**. We evaluate our method on two fundamental tasks: single-view forward rendering and novel view relighting from single-image to Relightable 3D. We evaluate the consistency between the rendered outputs and the ground truth reference images. The former involves single-view forward rendering with input G-buffers (such as normals, material, and depth information), image reconstruction from a single-image under random lighting, and relighting of a single image under unknown lighting. For single-view forward rendering, we compare against recent state-of-the-art neural rendering methods RGB↔X Zeng et al. (2024b), neural-gaffer Jin et al. (2024), DiLightNet Zeng et al. (2024a), and Diffusion-render Liang et al. (2025). For novel view relighting, we compare against recent open-source methods that support single-image to 3D generation with PBR materials, including Huyuan3D-2.1 Zhao et al. (2025), MeshGen Chen et al. (2025), 3DTopia-XL Chen et al. (2024), and SF3D Boss et al. (2024).

**Evaluation metric**. We use PSNR, SSIM Wang et al. (2004) and LPIPS Zhang et al. (2018) to measure the quality of the rendering.

**Evaluation datasets**. We randomly select 800 objects from the training data to create a test set, ensuring that these objects were not seen by the model during training. To validate the generalizability of our method, we utilize the Aria Digital Twin (ADT) Pan et al. (2023) and Digital Twin Catalog (DTC) Dong et al. (2025) datasets as out-of-domain datasets. These datasets provide comprehensive resources for 3D object modeling, featuring a vast library of highly detailed, photorealistic models with sub-millimeter accuracy. We further incorporate the Glossy Synthetic dataset Liu et al. (2023), which provides 3D assets, and expand it with additional assets sourced from the BlenderKit [1]. We also modify rendering nodes to utilize the Principled BSDF shader [2].

## 5.2 SINGLE-VIEW FORWARD RENDERING

**G-buffers forward rendering**. As shown in Fig. 5, we compare against Diffusion Renderer using ground truth G-buffers and LH-Slat (with Base Color SLAT), bypassing the single-image-to-intermediate representation step. Our method demonstrates superior accuracy in shadow and highlight distribution (e.g., the toy's specular highlight and the sculpture's shadow detail), likely due to our explicit 3D structural information. Furthermore, we accurately capture material reflections of ambient light, as illustrated by the stainless steel. Quantitatively, our method significantly outperforms baselines across four datasets in Fig. 6.

---

[1]https://www.blenderkit.com/

[2]https://www.blender.org/

Table 2: Ablation study on the number of blocks for $\mathcal{D}_E$.

| Num | PSNR | SSIM | LPIPS | Param. | FPS |
|---|---|---|---|---|---|
| 1 | 31.56 | 0.9608 | 0.0508 | 12.65M | 48 |
| 3 | 32.35 | 0.9635 | 0.0474 | 31.55M | 38 |
| **6** | 32.54 | 0.9649 | 0.0442 | 59.8M | 30 |
| 9 | 32.56 | 0.9645 | 0.0439 | 88.23M | 23 |

Table 3: Ablation study on decoder input types.

| Input types | PSNR | SSIM | LPIPS |
|---|---|---|---|
| base color | 30.38 | 0.9541 | 0.0564 |
| LH | 32.02 | 0.9631 | 0.0494 |
| LH + base color | 32.54 | 0.9649 | 0.0442 |

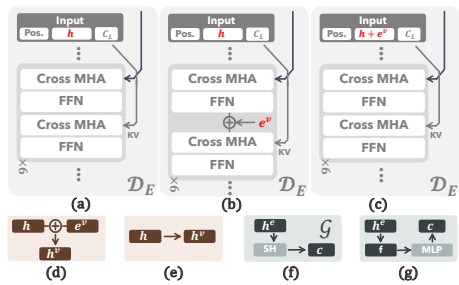

Figure 8: Different component designs for the feedforward network $\mathcal{D}$.

Table 4: Performance Comparison of Different Architectures.

| Arch | PSNR | SSIM | LPIPS |
|---|---|---|---|
| a + e + f | 29.82 | 0.9472 | 0.0642 |
| a + e + g | 30.66 | 0.9524 | 0.0515 |
| a + d + g | 31.96 | 0.9597 | 0.0492 |
| b + d + g | 32.43 | 0.9628 | 0.0472 |
| c + d + g (ours) | 32.54 | 0.9649 | 0.0442 |

**Random-lit single-image reconstruction**. As shown in Fig. 6, our method provides improved image reconstruction compared to the baseline and outperforms the Diffusion Renderer's estimated intrinsic property approach. Quantitative evaluations in Tab. 1 demonstrate our method's advantage across the majority of metrics.

**Unknown-lit single-image relighting**. Our method achieves more accurate highlights and color in relit images with unknown lighting, compared to other methods, as shown in Fig. 7. For example, observe the highlights on the speaker cones (first row) and the teapot color (second row). Tab. 1 further demonstrates our method's improved generalization performance

**Novel-view Relighting**. We compared our method for novel view relighting and reconstruction against state-of-the-art image-to-3D methods supporting PBR materials. Given a single image, we generate a 3D Gaussian or Mesh and use neural rendering for relighting, while other methods reconstruct the 3D model and use Blender. Fig. 1 shows that, with the same mesh, our method achieves more accurate lighting and material interactions than Hunyuan3D. Our quantitative results in Tab. 1 demonstrate significant improvements over other 3D generation methods.

### 5.3 ABLATION STUDY.

We perform ablation studies on our test set, investigating the depth of $\mathcal{D}_E$, input types, and network architecture. Increasing the number of layers in $\mathcal{D}_E$ improves performance but decreases inference speed; we select 6 layers for a balance between performance and accuracy (Tab. 2). LH-SLAT, containing richer information, performs better than base color SLAT alone; however, base color SLAT complements LH-SLAT, further improving performance when used together (Tab. 3). Providing view information early, followed by lighting information, ensures better performance and generalization for $\mathcal{D}_E$ (Fig. 8, Tab. 4).

### 6 CONCLUSION

We propose a compact multi-stage framework for relightable 3d generation, enabling consistent high-fidelity reconstruction and realistic relighting. Experiments show improved quantitative and perceptual results over strong baselines, and ablations confirm each component's contribution. Although evaluated on controlled captures with moderate compute, the approach suggests clear directions for in-the-wild and dynamic scenes and for efficiency and generalization improvements. We hope this work advances practical neural relighting and reconstruction.

ETHICS STATEMENT.

This work does not involve human subjects, personally identifiable information, or sensitive data. The datasets used in this study are publicly available and widely adopted in the machine learning community. All experiments were conducted using standard computational resources without environmental or societal harm. The methodology does not introduce discriminatory biases, and the model's potential applications are aligned with responsible AI principles. The authors have reviewed the ICLR Code of Ethics and confirm that this submission adheres to its guidelines.

REPRODUCIBILITY STATEMENT.

For reproducibility, we provide a comprehensive description of our model architecture, training, and evaluation in the main paper. Further implementation details, including data preprocessing and hyperparameters, are available in the Appendix. We aim to enable independent replication of our results through clear and thorough documentation

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

## A APPENDIX

### A.1 MORE DETAILS.

**Training details**. The training pipeline is executed on 4 NVIDIA H100 80GB HBM3 GPUs. The initial stage involves training the flow model, where we employ LoRA initialized using the PEFT Mangrulkar et al. (2022). The LoRA configuration consists of a rank of 512 and a scaling factor of 512. LoRA is applied to the query, key-value, output projection, and the combined query-key-value modules within the attention mechanism. The AdamW optimizer Loshchilov & Hutter (2019) is used with a learning rate of $1.0 \times 10^{-4}$. The first stage requires approximately one day for completion. In the second stage, we utilize the AdamW Loshchilov & Hutter (2019) optimizer with a batch size of 48 and a linear warmup learning rate of $1.0 \times 10^{-4}$ over 5,000 steps, followed by a cosine decay schedule. An end-to-end joint training of the IAD, EAR, and $\mathcal{E}_l$ is performed. Training acceleration is achieved through the implementation of Flash-Attention 3 Shah et al. (2024) and the gsplat Ye et al. (2025). Initially, the model is trained with all loss components for 400K iterations, requiring approximately 8 days. Subsequently, the PBR rendering loss is removed, and training continues for an additional 100K iterations, taking approximately 2 days.

## B LLM USAGE

We acknowledge large language models (LLMs) in the preparation of this manuscript. Specifically, we utilized LLMs for text polishing, grammar correction, and improving the clarity. The core experimental results and scientific contributions remain entirely our own work.

