# OpenReview forum: "RelitTrellis: Lightning-homogenized Structured 3D Latents For Relightable 3D Generation"
_ICLR.cc/2026/Conference — ICLR 2026 Conference Withdrawn Submission_

### Official Review · Reviewer_rJL4 · 2025-10-30

**Soundness:** 3
**Presentation:** 3
**Contribution:** 3
**Rating:** 4
**Confidence:** 5

**Summary:**

This paper proposes a two-stage pipeline which takes as input a single unknown lit render (image) of a 3D asset and outputs a screen space relightable GS (Gaussian splat) representation.
In the first stage a modified variant of SLAT titled LH-SLAT (Lighting Homogenized) is diffused while conditioned on the input image. The main proposed difference between a normal SLAT and an LH-SLAT is that the LH-SLAT contains canonical homogenized lighting (uniform white illumination).
In the second stage a feed-forward Transformer based module is designed and trained to map the LH-SLAT, a target HDRI lightmap and target camera view into the GS.

**Strengths:**

1. The problem definition is clearly explained.
2. The design of the 2nd stage feed-forward network is clearly motivated and presents some interesting technical details.
3. The experimental evaluation has succinct coverage.

**Weaknesses:**

**Major concerns**:
1. Although the idea of the LH-SLAT (1st claimed contribution), is verbally motivated, and clearly explained, it is not empirically justified. It is imaginable that SLAT representation with canonicalized lighting should aid the 2nd stage while compared to a regular SLAT with the leaked lighting, but since the stage 2 has all learned components, a strong counter argument can also be made that with sufficient training the 2nd stage should also be able to robustly learn the lighting canonicalization along with the relighting. Without comparing to this strong first baseline, it is very hard to asses how much improvement the LH-SLAT produces.

**Minor concerns**:
1. I suppose that for for the Image conditioned generation of the LH-SLAT representation (Figure 4.), both the SSFLOW and the SLAT-FLOW transformers of Trellis are used. But it would be helpful if clarified explicitly. Otherwise it's unclear how the sparsity pattern of the Sparse Voxels is obtained just from the input (unknown lit) image.
2. Equation 7 should explicitly show the conditioning on the image I_in, otherwise it confuses the method.
3. The distribution of the figures for LH-slat generation (sec 4.1) over Fig 3. and Fig 4. is slightly confusing. The exposition could benefit from making two separate figures for sec. 4.1 (showing the extraction as well as the diffusion) and a second figure for section 4.2.
4. Are the register tokens in IAD added for scenes with multiple objects? The paragraph L270-283 gives some information, but it would be helpful to clarify in detail. I can imagine that in case of multiple objects in the scene, the register tokens could enable message passing between the SLAT sequences of different objects, but not clear where they would be useful for the case of a singular object.
5. Section 4.2.2 title should be environment aware renderer*.
6. Why are the ray directions converted to camera coordinate system? By doing so, the ray-directions essentially only represent the pixel information, and their positional encodings of the directions would be same irrespective of the cameras. Clarification would be helpful.

**Questions:**

I am overall inclining towards accepting, but as I mentioned above, the major concern about whether the LH-SLAT representation is justified in this pipeline needs to be clearly shown empirically. That is the only reason keeping me from recommending an accept at this stage, and thus would be interested in the author's response before making the final decision.

---

### Official Review · Reviewer_dQHd · 2025-10-31

**Soundness:** 3
**Presentation:** 2
**Contribution:** 3
**Rating:** 8
**Confidence:** 5

**Summary:**

This paper tackles the task of single-image to relightable 3D generation. The authors propose a homogenize-then-synthesize pipeline with the argument that a homogenized space, i.e., a space where shadows and specularities are removed while diffuse cues are maintained, is better suited for relighting. For this, they train a flow-matching model that maps the input image to a homogenized feature space. Specifically, they render 3D assets with a white environment map and use TRELLIS's encoding mechanism to obtain the homogenized latents. Later, a LoRA is applied to the pretrained TRELLIS encoder to map the original TRELLIS latent to this new homogenized latent.

Next, they take the homogenized latent as input and produce intrinsic features. To further enhance view-dependent effects, the author proposes to use feature-based 3DGS rendering. Concretely, they use a model to take in the intrinsic feature, the encoded target lighting map, and the viewing direction as input to produce 3DGS attributes. Finally, a light MLP outputs RGB based on the generated attributes for 3DGS.

Experiments on various datasets verify the effectiveness of the proposed approach.

**Strengths:**

- originality-wise: the idea of homogenized space is interesting and seems effective. The idea of feature-based 3DGS rendering is also promising;
- quality-wise: the quantitative results demonstrate the effectiveness;
- clarity-wise: good in general but can be further improved;
- significance-wise: single-image to 3D religthing is an important task for various downstream applications, e.g., AR/VR.

**Weaknesses:**

## 1. About LH-SLAT effectiveness

Can authors clarify:

a. whether `base color` in L248 means the original/non-homogenized rendered images;

b. whether all models in Tab. 3 received the same amount of training under the same setup.

## 2. About the necessity of the intrinsic aware decoder (IAD)

I am curious whether this decoder is indeed needed. This decoder seems to make the whole pipeline quite complicated and does not provide too much information, as the homogenized latent should contain the intrinsic information already. Can authors try to see how severe the performance will drop if we remove IAD?

Further, can authors clarify whether the loss of material properties (L358) is imposed on the decoder for the intrinsic features?

## 3. Qualitative results

There are too few qualitative results in the paper. I am not fully convinced by numbers only, especially for relighting tasks. Please provide more qualitative results and videos to demonstrate the performance fully.

## 4. References

- Please change `\cite` to `\citep`

- missing related works for generative 3D relighting

[a] Ginter et al., A Diffusion Approach to Radiance Field Relighting using Multi-Illumination Synthesis. EGSR 2024.

[b] Zhao et al., IllumiNeRF 3D Relighting without Inverse Rendering. NeurIPS 2024.

**Questions:**

See "Weakness"

---

### Official Review · Reviewer_oqG3 · 2025-11-01

**Soundness:** 3
**Presentation:** 3
**Contribution:** 3
**Rating:** 6
**Confidence:** 2

**Summary:**

This paper presents RelitTrellis, a two-stage framework for single-image relightable 3D generation. It first extracts a Lighting-Homogenized Structured 3D Latent (LH-SLAT) that removes shadows and highlights while retaining geometry-consistent cues, then decodes it into a relightable 3D Gaussian Splatting field conditioned on target lighting. The method achieves real-time inference, strong cross-view consistency, and state-of-the-art performance on multiple benchmarks. Overall, it offers a principled and efficient alternative to traditional PBR decomposition and diffusion-based relighting.

**Strengths:**

- The paper introduces a well-motivated homogenize-then-synthesize design that effectively stabilizes relightable 3D generation from a single image. The proposed lighting-homogenized latent improves disentanglement of geometry and illumination, enhancing controllability and consistency.
- The method is tested on diverse benchmarks (Digital Twin Category, Aria Digital Twin, Objaverse, and a Glossy Synthetic set), with a rigorous array of baselines. Table 1 demonstrates consistent advantages in LPIPS/PSNR/SSIM across multiple sub-tasks, and ablations in Tables 2-4 further clarify each component’s value.
- The feed-forward 3D Gaussian decoder supports real-time inference while maintaining high-quality relighting.

**Weaknesses:**

- The majority of the experiments are performed on synthetic datasets with known geometry and materials. While the Digital Twin Category and Aria Digital Twin sets represent realistic assets, the assumption that single images are rendered with clean backgrounds and known object-centric crops somewhat limits real-world applicability. There are no tests on “in-the-wild” photos, complex backgrounds, or non-curated internet data, making true generalization less certain.
- The paper could be improved by directly analyzing situations where the method fails: e.g., highly specular or transparent objects (where lighting homogenization may destroy necessary detail), or when the input image is heavily occluded or corrupted. Figure legends only highlight successes.
- While Table 3 shows performance with different input types, further ablations on LH-SLAT extraction mechanisms (e.g., different flow architectures, varying degrees of shadow/highlight suppression, full vs. partial homogenization) would provide more insight into where the performance gains genuinely originate.

**Questions:**

- Can the method handle in-the-wild images with complex global illumination or background clutter? How well does the model generalize to non-object-centric or dynamic scenes beyond the tested datasets?
- How does performance vary across material classes (highly glossy/metallic, translucent) and does the shadow/highlight handling degrade for extreme specularities? Any per-category breakdown?
- There are a few cases in Tab. 1 where the proposed method does not outperform baselines. Can you provide an intuitive explanation of why that happens? What could be wrong with those models and specific datasets?

---

### Official Review · Reviewer_tNsp · 2025-11-03

**Soundness:** 1
**Presentation:** 2
**Contribution:** 2
**Rating:** 2
**Confidence:** 3

**Summary:**

The paper proposes RelitTrellis, a two-stage pipeline for single-image relightable 3D generation. Stage 1 performs lighting homogenization to produce a “Lighting-Homogenized Structured 3D Latent (LH-SLAT)” that removes shadows and specularities. Stage 2 decodes this latent into a relightable 3D Gaussian field using an Intrinsic-Aware Decoder and an Environment-Aware Renderer. Experiments claim improved fidelity and multi-view consistency over diffusion-based or inverse-rendering baselines.

**Strengths:**

- The writing is generally easy to follow, though the complexity of the system makes things a bit harder to digest.

- This work addresses an important and difficult problem: single-image relightable 3D reconstruction. It attempts to unify physically-interpretable and feed-forward generative paradigms.

- Quantitative results are reported on several benchmarks (ADT, DTC, Objaverse) and multiple relighting tasks.

**Weaknesses:**

- The qualitative results do not convincingly demonstrate superiority. In Figure 1, several relit examples not convincingly better than the baselines: the first row shows incorrect specular highlights and the third row shows color shifts inconsistent with the ground truth.

- Also, I could not seem to locate video comparisons in a supplementary. Such video comparisons are very important for visual judgment of relighting consistency and specularity correctness.

- The method involves multiple coupled networks and hand-crafted training stages—a rectified-flow LH-SLAT extractor, LoRA-based adaptation, a transformer-style intrinsic decoder, and a cross-attention lighting renderer—each with its own training losses and data generation pipeline. Without an official code release (and the authors do not seem to commit to one), reproducibility might be doubtful.

- The “lighting-homogenization” step removes illumination cues before the relighting decoder. However, specularities, glossiness, and other material-dependent signals are precisely what guide relighting. By suppressing them early, the pipeline may discard useful information and force the later decoder to re-infer material cues from the lighting-harmonized latents—potentially an intrinsically ill-posed design.

**Questions:**

I don't have further questions.

---

### Note · Authors · 2025-11-14

**Comment:**

Thank you to the reviewers for their valuable comments; we will make detailed revisions and additions.

**Withdrawal Confirmation:**

I have read and agree with the venue's withdrawal policy on behalf of myself and my co-authors.